# Central Sensitization Syndromes and Trauma: Mediating Role of Sleep Quality, Pain Catastrophizing, and Emotional Dysregulation Between Post-Traumatic Stress Disorder and Pain

**DOI:** 10.3390/healthcare13172221

**Published:** 2025-09-04

**Authors:** Elena Miró, Ana Isabel Sánchez, Ada Raya, María Pilar Martínez

**Affiliations:** 1Department of Personality, Assessment and Psychological Treatment, University of Granada, 18071 Granada, Spain; aisabel@ugr.es (A.I.S.); mnarvaez@ugr.es (M.P.M.); 2Mind, Brain and Behavior Research Center, University of Granada, 18071 Granada, Spain; adapsicodolor@gmail.com

**Keywords:** trauma, post-traumatic stress disorder, central sensitization syndromes, sleep quality, pain catastrophizing, emotional dysregulation

## Abstract

**Background:** Central sensitization syndromes (CSSs) are associated with a high incidence of traumatic events; however, few studies have examined the potential mechanisms linking post-traumatic stress disorder (PTSD) and pain. **Objectives:** The present research aims to clarify this association by exploring the presence of trauma, PTSD, and related clinical variables in participants with CSSs compared to healthy controls and those with medical problems. **Methods:** A large sample of both sexes of the Spanish general population (*n* = 1542; aged 18–84 years) completed an online survey assessing the presence of traumatic experiences (psychological trauma, physical trauma, physical and sexual abuse), PTSD, and other clinical measures (central sensitization, pain, sleep quality, anxiety, depression, perceived stress, and emotional regulation). **Results**: The CSS group (*n* = 467) showed a higher incidence of repeated trauma, PTSD, and dissociative symptoms compared to the medical pathologies (*n* = 214) and healthy (*n* = 861) groups. The CSS group also showed greater clinical impairment than the other groups, especially the CSS subgroup with PTSD. In this subgroup, PTSD symptoms were significantly correlated with the remaining clinical measures, and sleep dysfunction, pain catastrophizing, and emotional dysregulation mediated the relationship between PTSD and pain, accounting for 55.3% of the variance. **Conclusions:** CSS represents a major therapeutic challenge. An integrated treatment addressing both trauma and pain, with an emphasis on sleep quality, pain catastrophizing, and emotional regulation, could improve the effectiveness of the current therapeutic approaches.

## 1. Introduction

In the field of chronic pain, defined as pain persisting for more than 3 months [1], there is an important group of conditions where medical tests fail to detect abnormalities that can be demonstrated to be their cause. These pain disorders have been referred to by various terms, including unexplained chronic pain syndromes or nociplastic pain, although “central sensitization syndromes” (CSSs) remains the most widely used [2,3]. CSSs constitute a major global health concern, affecting up to 30% of the population [4], resulting in healthcare costs that exceed EUR 12 billion in developed countries and huge indirect costs due to loss of productivity and reduced quality of life [5].

CSSs include fibromyalgia, chronic fatigue syndrome, multiple chemical sensitivity, irritable bowel syndrome, tension headaches and migraines, and most cases of temporomandibular, pelvic, abdominal, and chronic back pain [2,3]. These conditions show high comorbidity and share, in addition to pain, the presence of sleep disturbances, cognitive–affective alterations, and significant life stress, which has led to the suggestion that they are a common condition with different clinical manifestations [2,6]. Although the pathogenesis of CSSs remains unclear, a state of central sensitization (CS) is observed, defined as increased responsiveness of nociceptive neurons to normal or subthreshold afferent input, which produces hyperactivity in the pain neuromatrix and dysfunction of descending pain inhibition, leading to generalized hyperalgesia and allodynia [2,3,7].

Between 50 and 90% of people with CSSs experience sleep disorders, especially those with fibromyalgia (FM), where the rates reach between 94.7% and 99% [8,9]. In addition to poor subjective sleep quality, objective polysomnography studies have reported a reduction in total sleep time, more awakenings and arousals, increased onset latencies of sleep, more phase changes and phase 1 of light sleep, a lower percentage of slow wave sleep, and abnormalities indicative of slow wave sleep instability, such as alpha–delta sleep, which seem to be associated with a state of hyperarousal prior to sleep that fragments sleep [8,9,10]. The coexistence of these sleep alterations and pain leads to greater pain intensity, fatigue, cognitive difficulties, emotional distress, and disability [8,11]. Pain and sleep are bidirectionally related, with poor sleep quality being a more powerful predictor of pain in longitudinal research than the other way around [10,12].

On the other hand, people with CSSs show cognitive–affective characteristics that may predispose them to or maintain CSSs [6,13]. One of the most studied aspects is pain catastrophizing [14], defined as the tendency to magnify the threat value of pain, ruminate about it, and feel helpless in relation to pain. It has been demonstrated using the fear-avoidance model [15], which influences pain intensity and pain-related disability across different clinical populations [14,16]. Furthermore, at least a quarter of CSS patients have depression or an anxiety disorder [13,17], and the presence of symptoms is much higher (with up to 87.5% and 76.1% showing anxiety and depressed mood, respectively) [18]. Emotional distress exacerbates pain and associated disability [16,19], and it can precede and/or follow pain, although depression tends to follow pain while anxiety precedes pain onset in up to 75% of cases [13].

It has recently been highlighted that the presence of trauma is higher among CSSs patients than healthy controls and other chronic pain patients [20,21]. FM is the most studied condition, with a trauma rate of between 20 and 70% of cases, most studies have found a link between physical (especially psychological) trauma and FM onset [22,23]. However, there are few studies on this topic, and those that are available are of low quality (i.e., many do not include control groups or combine medical pain problems with CSSs); additionally, most studies have explored only FM, while others have defined the type of trauma inconsistently (see for a review [20,21]). It is recommended to distinguish between physical and psychological trauma, which, in addition to emotional trauma (loss or relational conflicts), includes sexual abuse (any unwanted sexual contact), physical abuse (any contact that causes pain), and emotional abuse (verbal abuse, neglect, making someone feel worthless, and controlling their behavior) [23].

Studies analyzing the relationship between different types of trauma and pain or other health measures have generated inconsistent findings, with some studies finding significant relationships between physical, sexual, or emotional abuse and somatic symptoms or psychological state [24,25], and others finding no relationship between these factors [26,27]. However, studies that explore associations between PTSD diagnostic criteria with FM [18] or other chronic pain conditions [28,29] clearly show that the impact of trauma is up to four times greater than the trauma itself. The mechanism linking trauma and CS remains unclear, but it has been postulated that it is through the development of PTSD. Haüser et al. found that post-traumatic stress symptoms (PTSSs) mediated the relationship between traumatic events and widespread pain in the general population [30]. McKernan et al. highlighted the mediating role of PTSD between trauma and chronic pain in interstitial cystitis/bladder pain syndrome [29]. In addition, PTSD/PTSSs tended to precede pain in studies that analyzed the temporal relationships between these factors [18,31,32].

The mechanisms that connect PTSD/PTSSs with pain are unknown. One study found that the relationship between pain and PTSSs occurring after sexual abuse was mediated by depression [33]. However, the data obtained by Häuser et al. did not support the conclusion that the relationship between PTSSs and FM is mediated by depression [30]. Miró et al. found that anxiety played a mediating role in the relationship between PTSSs and FM status [32], and Kascakova et al. found that insecure attachment mediates the relationship between childhood trauma and migraine, although the presence of PTSD/PTSSs was not explored [34]. Although preliminary research has highlighted the association between PTSD and pain [21,35], more research is needed to understand the association.

This study aimed to determine the presence and characteristics of traumatic experiences and PTSD in the general population, comparing participants with CSSs with those with medical problems (MPs) and healthy controls. Furthermore, the present study compared the main clinical characteristics relevant to pain problems in the CSS, MP, and healthy groups and analyzed whether the CSS subgroup with PTSD was more clinically affected than the CSS subgroup without PTSD, exploring the relationships between PTSD and clinical measures and examining possible mediators of the relationship between PTSD and pain. We hypothesized that the presence of PTSD was higher in the CSS group than in the other groups. The CSS group showed greater clinical impairment in terms of the evaluated symptoms, especially the CSS subgroup with PTSD; in addition, anxiety or aspects linked to it will be identified as mediators between PTSD and pain.

## 2. Materials and Methods

### 2.1. Participants and Procedure

The sample comprised 1542 Spanish adults from the general population (1058 women, 471 men, and 13 non-binary individuals) aged 18–84 years (*M* = 31.80 years, *SD* = 15.15). This sample was divided into 3 groups: participants without any clinical conditions (medical or psychological), classified as the healthy group (*n* = 861); participants with one or more conditions associated with CSSs, classified as the CSS group (*n* = 467); and participants with a medical pathology, classified as the MP group (*n* = 214) (for more details, see Appendix A).

Participants completed an online survey (lasting approximately 30 min) administered via the LimeSurvey platform (https://www.limesurvey.org/es/, assessed from November 2023 to March 2024). The survey was disseminated through the institutional mail of the University of Granada, social media, and community contexts. In addition, pain-related support groups were invited to participate in the survey. The study was approved by the Ethics Committee of the Andalusian Regional Government (Spain) (1922-N-19, approved on 25 November 2019). All participants provided signed informed consent to participate in the study. The inclusion criteria required participants to be 18 years or older and be willing to complete the survey voluntarily without any reward. The first section of the survey collected information on sociodemographic data (age, gender, marital status, children, education level, and employment status) and health data (diagnosed diseases, subjective health status, medication consumption, surgical operations, and, for female participants, whether they were currently in any phase of menopause). The second section consisted of several questions about traumatic experiences and the presence of PTSSs/PTSD, CS, characteristics of pain and pain catastrophizing, sleep quality, anxiety and depression, perceived stress, and emotional regulation.

### 2.2. Measures

#### 2.2.1. Traumatic Events and PTSSs/PTSD

An interview was developed to assess the presence of trauma, PTSSs, and PTSD according to the Diagnostic and Statistical Manual of Mental Disorders (DSM-5) [36] and based on the Global Assessment of Post-traumatic Stress (GAPS) [37]. The GAPS is the most recent scale developed and validated in a Spanish population exposed to various traumatic events; it facilitates the evaluation of PTSSs and the diagnosis of PTSD, including all DSM-5 criteria at the population level, with easy and brief application [37]. The GAPS has an internal consistency of α = 0.91 and good discrimination (g = 1.27) and convergent validity (r = 0.78) [37]. It is composed of 58 items divided into 3 sections that cover the evaluation of traumatic events, symptomatology, and the individual’s functioning. Our final interview was composed of two sections. In the first part, the list of traumatic events included in Section 1 of the GAPS was extended from 11 to 16 in order to collect specific aspects of sexual abuse and physical or psychological abuse that are relevant in this research. For each of these items, the person provides answers based on the possibilities of experiencing a trauma included in the DSM-5: “I have not experienced it”, “I had a direct experience”, “I witnessed the event that happened to someone else”, or “It happened to someone close to me” (Criterion A). We then added questions to which the person had to briefly explain the most important traumatic event, indicate the age at which the event occurred, and state how often the event occurred (once, on several occasions, or repeatedly). In the second part, we included items to evaluate the presence of intrusive symptoms (Criterion B), avoidance of stimuli associated with the event (Criterion C), alterations in cognition and mood (Criterion D), hyperarousal (Criterion E), and impairment of daily functioning (Criterion G). There were a total of 28 items scored from 0 to 3 points (*not at all*, *a little, quite a lot,* and *a lot*). Questions on symptom duration (Criterion F), time of onset, and presence of dissociative symptoms were also included. The presence of PTSD symptoms was determined by categorizing the Likert-type responses (“not at all,” “a little,” “quite a lot,” or “a lot”) to the items corresponding to the symptoms as “symptom not present” (if the response was “not at all”) and as “symptom present” (if the response was any of the remaining options: “a little,” “quite a lot,” or “a lot”). PTSD diagnosis required the presence of at least one symptom in Criteria B and C to occur for at least one month (Criterion F), and at least two symptoms in Criteria D and E together with clinically significant interference with daily life (Criterion G). No cut-off scores were applied to allow for greater flexibility and nuance in data interpretation, and because there are no validated cut-off scores with appropriate sensitivity and specificity for the target population. Three independent expert evaluators conducted an initial assessment of the interview questions to determine their suitability. In this sample, the Cronbach’s alpha coefficients for the subscales were as follows: intrusive symptoms (0.89), avoidance behaviors (0.83), cognitive and mood symptoms (0.90), hyperarousal symptoms (0.86), dissociative symptoms (0.88), and interference with daily life (0.90).

#### 2.2.2. Central Sensitization

CS was evaluated using the Central Sensitization Inventory (CSI) [38], which has two sections: Part A assesses 25 somatic and emotional health-related symptoms using a 5-point Likert scale, from *never* to *always*, resulting in total scores ranging from 0 to 100, where 40 is the indicative cut-off point for CSSs. Part B consists of a list of 10 diseases primarily associated with CSSs and is not scored. The CSI has strong psychometric properties, including high test–retest reliability, internal consistency, construct validity, criterion validity, and cross-cultural validity, making it a useful tool for identifying individuals with central sensitization in populations suffering from chronic pain [39]. The Spanish version of the CSI has high internal consistency (*α* = 0.87) and test–retest reliability (0.91) [40]. In this sample, the Cronbach’s alpha coefficient was 0.91.

#### 2.2.3. Pain

The McGill Pain Questionnaire–Short form (MPQ-SF) [41] assesses several pain dimensions using 15 pain descriptors (sensory and affective), with 4 response options (from *no* to *severe*), a current pain index (ranging from 0, *no pain at all*, to 5, *insufferable pain*), and a visual analogue scale (VAS) to evaluate pain intensity during the previous week (from 1, *no pain,* to 10, *extreme pain*). The present study used the VAS and the sensory–affective scale. The Spanish version has appropriate internal consistency (*α* = 0.74) and discriminant validity [42]. In this sample, the Cronbach’s alpha coefficient for the sensory–affective scale was 0.91.

#### 2.2.4. Pain Catastrophizing

The Pain Catastrophizing Scale (PCS) consists of 13 items that evaluate magnification, rumination, and helplessness [43]. Items are assessed from 0, *not at all*, to 4 *all the time*, and the total scale score ranges from 0 to 52 points. The Spanish version of the PCS showed good internal consistency (*α* = 0.79), test–retest reliability, and sensitivity to change [44]. In this study, the Cronbach’s alpha coefficient for the PCS was 0.95.

#### 2.2.5. Sleep Quality

Sleep quality was assessed using the Pittsburgh Sleep Quality Index (PSQI) [45]. The PSQI includes 19 items grouped into seven dimensions: subjective sleep quality, sleep latency, sleep duration, sleep efficiency, sleep disturbances, use of sleeping medication, and daytime dysfunction. The subscale scores range from 0 to 3, and the total score ranges from 0 to 21 (higher scores indicate greater sleep disturbance). The Spanish version shows adequate internal consistency (*α* = 0.80), test-retest reliability, and convergent validity [46]. The Cronbach’s alpha in the present study was 0.81.

#### 2.2.6. Anxiety and Depression

Anxiety and depression symptoms were assessed using the Hospital Anxiety and Depression Scale (HADS) [47]. The HADS comprises 14 items (rated from 0 to 3 points) divided into two subscales, anxiety and depression, with scores ranging from 0 to 21 for each subscale; scores of 11 or more were considered clinically relevant. The Spanish adaptation has good internal consistency (*α* = 0.86) and test–retest reliability [48]. In this sample, the Cronbach’s alpha coefficients for the anxiety and depression scales were 0.84 and 0.80, respectively.

#### 2.2.7. Stress

The Perceived Stress Scale (PSS) was used to assess the level of perceived stress in the previous month [49]. The scale comprises 14 items, rated from 0 (*never*) to 4 (*very often*), designed to assess how unpredictable, uncontrollable, and overwhelming the respondent perceives their life to be. The total scores range from 0 to 56. The Spanish adaptation shows adequate internal consistency (*α* = 0.81) and test–retest reliability, as well as concurrent validity and sensitivity [50]. The Cronbach’s alpha for the present study was 0.87.

#### 2.2.8. Emotional Processing

The Emotional Processing Scale-25 (EPS-25) was used to evaluate difficulties in emotional regulation [51]. It includes 25 items that measure the suppression of negative emotional states, unprocessed emotions, uncontrollability, avoidance, and disconnection from emotions, with ratings ranging from 0 (*completely disagree*) to 9 (*completely agree*); higher scores indicate greater difficulties with emotional regulation. The Spanish adaptation has an acceptable internal consistency (*α* = 0.91) and test–retest reliability [52]. In this sample, the Cronbach’s alpha coefficient was 0.89.

### 2.3. Data Analysis

The SPSS-28.0.1.0 software was used to conduct comparative and correlation analyses, and the JASP 0.17.2 software was used for mediation analyses. The total sample and the different subgroups analyzed were well above the minimum required sample size (*n* = 172), assuming an anticipated effect size of 0.5, a desired statistical power of 0.90, and a significance level of 0.05. Chi-square (*χ*^2^) tests and/or one-factor ANOVA were used to compare sociodemographic/clinical characteristics between subgroups (CSSs, MP, and healthy). Levene’s test was applied to determine the homogeneity of variance. For post hoc contrast, the Tukey test and/or Tamhane’s T2 test were used. Student *t*-test for independent samples was used to compare clinical variables between CSS participants with PTSD (*n* = 128) vs. CSS participants without PTSD (*n* = 127). Effect sizes were calculated using Cohen’s *d* and *ηp*^2^ (small effect size: *d* = 0.2 to 0.4, and *ηp*^2^ = 0.01 to 0.039, medium: *d* = 0.5 to 0.7, and *ηp*^2^ = 0.06 to 0.11, and large: *d* > 0.8, and *ηp*^2^ > 0.14).

In the CSS subgroup with PTSD, the relationship between clinical measures was explored using Pearson’s correlation coefficients. To reduce the probability of a Type I error, the Bonferroni correction was used. In this subgroup, a multiple mediation model was analyzed, considering as mediators between PTSD and pain, the measures most correlated with these variables and previous studies. The minimum number of samples required for regression analyses in the mediation model was *n* = 98 participants, considering an alpha level of 0.05, three mediators, a desired statistical power of 0.90, and an anticipated effect size of 0.15. Direct effects, indirect effects (specific and total), and total effects were estimated [53]. The direct path is the effect of variable X on variable Y. The specific indirect effect is the path linking X to Y via a specific mediator. The total indirect effect of X on Y is the sum of the specific indirect effects of all the mediators. The total effect of X on Y is the sum of the direct and indirect effects. When two or more variables are significant mediators between X and Y, indirect effects are significant in the mediation analyses. Delta method standard errors, bias-corrected bootstrap percentile confidence intervals, and ML estimator were used to test the significance of the effects. Estimates were based on 1000 bootstrap samples, and 95% confidence intervals were considered. Estimates of all paths were calculated using OLS regression.

## 3. Results

### 3.1. Sociodemographic Characteristics

The mean age of the healthy group (*M* = 28.33, *SD* = 13.04) was somewhat lower than that of the CSS group (*M* = 35.26 years, *SD* = 15.57) and the MP group (*M* = 38.20 years, *SD* = 18.10), *F*(2,1539) = 57.77, *p* < 0.001. Regarding gender, the CSS group had a significantly higher proportion of women (78.8% women, 21.2% men) compared to the MP group (58.9% women, 41.1% men) and the healthy group (66.6% women, 33.4% men), *χ*^2^(2) = 33.35, *p* < 0.001. In addition, there were significantly higher prevalences of married or cohabiting individuals in the CSS group (28.7% married, 60.2% single, 9.9% divorced, 1.3% widowed) and the MP group (35% married, 55.6% single, 8.9% divorced, 0.5% widowed) than in the healthy group (17.7% married, 78% single, 4% divorced, 0.3% widowed), *χ*^2^(8) = 75.61, *p* < 0.001. The education level of the healthy group was significantly higher (1.7% elementary education, 5.6% secondary education, and 92.7% vocational training or university education) than that of the CSS group (4.7%, 7.1%, and 88.2%, respectively) and the MP group (5.2%, 5.1%, and 89.8%, respectively), *χ*^2^(10) = 31.11, *p* < 0.001. Finally, as expected, there was a significantly higher prevalence of retired people or people on sick leave in the CSS group (83.7% active, 7.9% unemployed, 3.9% retired, 4.5% on temporary/permanent sick leave) and in the MP group (83.1% active, 4.2% unemployed, 8.4% retired, 3.2% on temporary/permanent sick leave) compared to the healthy group (94.4% active/studying, 3.9% unemployed, 1.5% retired, 0.2% on sick leave), *χ*^2^(10) = 105.92, *p* < 0.001. These data are similar to those described in studies of CSSs [4,18].

### 3.2. Frequency and Characteristics of Traumatic Events

The frequency of traumatic events was significantly higher in the CSSs (54.6%) and MP (54.7%) groups than in the healthy group (41.6%), *χ*^2^(2) = 25.96, *p* < 0.001. There were no significant differences in the time of trauma occurrence, *χ*^2^(4) = 5.70, *p* = 0.22. In all groups, trauma occurred more frequently in adolescence (38.5%, 28.2%, and 39.7% in the CSS, MP, and healthy groups, respectively) or adulthood (42.6%, 48.4%, and 42%) than in childhood (18.9%, 23.4%, and 18.2%). The presence of repeated trauma was higher in the CSS group (30%) than in the MP (19.4%) or healthy (21%) groups, *χ*^2^(4) = 11.33, *p* = 0.023. Furthermore, there was a significantly higher proportion of individuals who met the criteria for PTSD diagnosis in the CSS group (28.5%) than in the MP group (21.5%) and the healthy control group (15.6%), *χ*^2^(2) = 31.44, *p* < 0.001; additionally, the CSS group had a higher percentage of individuals with PTSD with dissociative symptoms compared to the other groups (19.5%, 12.1% and 8.9%, respectively), *χ*^2^(2) = 30.64, *p* < 0.001.

The most important traumatic event experienced directly by individuals in all groups was the sudden or accidental death of a loved one, with higher percentages in the CSS (33%) and MP (34.1%) groups than among healthy people (23.7%), *χ*^2^(6) = 23.28, *p* < 0.001. The frequencies of traumatic events were as follows: transport accidents (26.6%, 21%, and 18.2% in CSS, MP, and healthy groups, respectively), natural disasters (27%, 18.2%, and 17.8%), bullying/psychological abuse in the work place/academic area (26.3%, 19.2%, and 15.8%), bullying/psychological abuse in the intrafamilial/partner area (24.6%, 17.3%, and 13.6%, respectively), sexual abuse/rape by known people/strangers (21%, 10.3%, and 11%), and intrafamilial/couple physical abuse (16.3%, 8.9%, and 5%), in many cases with percentages almost double in the CSS group compared to the MP and healthy control groups (χ^2^(6) between 16.65, *p* < 0.05, and 61.42, *p* < 0.001). Other traumas occurred at low frequencies (see Appendix A).

### 3.3. Comparison of Clinical Characteristics in CSS, MP, and Healthy Groups

The three groups differed significantly in CS, pain intensity, sensory–affective pain, sleep quality, anxiety, depression, pain catastrophizing, perceived stress, emotional regulation, and PTSS, from *F*(2,1539) = 6.45, *p* < 0.01, *ηp*^2^ = 0.01 to *F*(2,1539) = 101.07, *p* < 0.001, *ηp*^2^ = 0.11. In post-hoc contrasts, the CSS group had greater overall clinical deterioration compared to the healthy group on all variables and compared to the MP group on all measures except for avoidance symptoms, negative cognitions and mood, and PTSD dissociative symptoms (where it showed greater clinical impairment, although it did not reach statistical significance) (see Table 1).

### 3.4. Comparison of CSS Participants with and Without PTSD

The subgroup of participants with CSS who did develop PTSD (*n* = 128), compared with CSS without PTSD (*n* = 127), showed significantly higher scores, indicative of clinical deterioration in terms of CS, pain intensity, sensory–affective pain, sleep quality, anxiety, depression, pain catastrophizing, perceived stress, and emotional regulation, from *t*(253) = 2.60, *p* < 0.01, *d* = 0.32, to *t*(253) = 8.27, *p* < 0.001, *d* = 1.03. The largest differences were observed in CS, anxiety, perceived stress, and emotional regulation (Cohen’s *d* between 0.80 and 1.03) (see Table 2).

### 3.5. Relationships Between Clinical Variables in the CSS Subgroup with PTSD

The different categories of PTSD symptoms show multiple correlations with the rest of the clinical measures collected in the study (see Table 3). The intensities of cognitive and mood symptoms, as well as hyperarousal, showed the greatest number of significant correlations with the remaining clinical symptoms, ranging from *r*(128) = 0.22, *p* < 0.0005, to *r*(128) = 0.60, *p* < 0.0005. The correlations that stand out are between hyperarousal and CS (*r*(128) = 0.60, *p* < 0.0005), hyperarousal and pain intensity or sensory–affective pain (*r*(128) = 0.42 and 0.46, *p* < 0.0005), hyperarousal and poor sleep quality (*r*(128) = 0.47, *p* < 0.0005), cognitive and mood symptoms of PTSD and hyperarousal with anxiety (*r*(128) = 0.48 and 0.53, *p* < 0.0005), hyperarousal and pain catastrophizing (*r*(128) = 0.47, *p* < 0.0005), hyperarousal an perceived stress (*r*(128) = 0.49, *p* < 0.0005), and hyperarousal and daily impairment with emotional regulation (*r*(128) = 0.53 and 0.47, *p* < 0.0005).

### 3.6. Mediators of the Association Between PTSD and Pain Intensity

Based on previous studies and the pattern of significant correlations observed, sleep quality, pain catastrophizing, and emotional regulation were examined as mediators of the effect of PTSD on pain intensity in the CSS subgroup with PTSD (see Table 4 and Figure 1). The specific indirect effects of sleep quality, pain catastrophizing, and emotional regulation, as well as the total indirect effect, were significant (between *z* = 2.60, *p* < 0.001, and *z* = 5.07, *p* < 0.001). However, the direct effect of PTSD on pain intensity did not reach statistical significance (*z* = −0.48, *p* = 0.631). The total effect, considering both direct and indirect effects, was significant (*z* = 3.72, *p* < 0.001). PTSD had a significant impact on pain intensity across all three mediating variables, which together explain 55.3% of the variance.

## 4. Discussion

This study aimed to explore the link between traumatic experiences and chronic pain, comparing participants with CSS to those with MP and healthy people from non-clinical settings. We found that the CSS and the MP groups had higher incidences of trauma compared to the healthy group (54.6%, 54.7%, and 41.6%, respectively). Previous studies have highlighted that experiencing traumatic events is associated with physical health problems [54], and the incidence of trauma has been reported to be higher in individuals with CSS than in healthy controls [20,21,22,28]. For example, in a German University outpatient clinic, Manuel et al. found an incidence of trauma of 54% in a CSS group [28], closely aligned with our results. In two studies in Spain, Miró et al. and Gardoki-Souto et al. found incidences of trauma of 75.2% and 84%, respectively, which were higher than those observed in our study, probably because they only included participants with FM, which may be the CSS condition with the highest rate of trauma [18,32]. Miró et al. also found a high trauma incidence of 52.9% in the healthy group without FM [32], while Gardoki-Souto et al. did not include a control group [18]. Additionally, comparison of the CSS group with the MP and healthy control groups, showed a higher frequency of recurrent trauma (30%, 19.4%, and 21%, respectively), higher rates of PTSD (28.5%, 21.5%, and 15.6%) and of PTSD with dissociative symptoms (19.5%, 12.1%, and 8.9%), as we hypothesized, consistent with studies that analyzed the presence of PTSD in individuals 2023with FM [18] or other chronic pain conditions [28,29]. PTSD estimates vary widely, depending on the type of CSS, the instrument used for assessment, and the context in which the samples are collected. A recent systematic review conducted by Karimov-Zwienenberg et al., which included 13 studies from different countries, mostly European, reported a PTSD prevalence of between 10.7% and 37% in individuals with CSSs, consistent with our findings [21]. Similarly, a previous meta-analysis of studies from different countries found a PTSD prevalence of 20.5% in widespread chronic pain compared to 5.1% in the general population [22].

The most important traumatic event experienced directly across all groups was the sudden or accidental death of a loved one, with higher percentages in the CSS (33%) and MP (34.1%) groups than in the healthy group (23.7%). This was followed, in descending order of frequency, by transport accidents, natural disasters, bullying/psychological abuse in the workplace/academia, in the intrafamilial/partner area, sexual abuse/rape by known people/strangers, and intrafamilial/couple physical abuse, with percentages often nearly twice as high in the CSS group as in the other groups. Previous research consistently indicated that emotional trauma—particularly the unexpected death of a loved one—was the most common trauma associated with CSS, followed by psychological abuse, with rates between 20 and 70% [18,21,23,25,28,32], which aligns with our percentages (24.6–33%) for these types of traumatic events.

In addition to emotional trauma, previous studies have also highlighted the high incidence of physical trauma, especially car accidents, followed by physical injury or surgery, with rates of 20–40% [18,28,32], which coincides with the rate of transport accidents observed in the CSS group (26.6%) in this study. Finally, the rates of physical (16.3%) and sexual abuse (21%) in the CSS group in our study are also noteworthy. Previous studies have reported higher incidences of physical abuse [32,55] and sexual abuse, especially in FM [18,30,33], which is the trauma with the highest risk of leading to PTSD [21].

Furthermore, although several studies have highlighted the association between childhood trauma and CSSs [18,20], we found that the traumas reported by the three study groups occurred more frequently in adolescence or adulthood than in childhood. This result seems consistent with the prevalence of lifelong victimization noted in other studies, with chronic stress being more prevalent than acute stress [21,25,31,32,55]. For example, Pierce et al. found that recent trauma in adulthood acts as a mediator linking trauma before 17 years of age to generalized sensory sensitivity, affective distress, and cognitive dysfunction in chronic pelvic pain [25].

Comparison of the clinical symptoms evaluated in the three study groups revealed higher levels of CS, pain, pain catastrophizing, sleep disruption, anxiety, depression, perceived stress, emotional dysregulation, and PTSS intensity in the CSS group compared to the other two groups, which is consistent with the distinct clinical aspects of CSSs [2,6]. In addition, compared to the CSS subgroup that did not develop PTSD, the CSS subgroup that developed PTSD showed greater clinical impairment for all clinical measures. In this subgroup, PTSSs showed various correlations with the clinical measures evaluated, with the association between cognitive and mood symptoms of PTSD with anxiety, especially the association between hyperarousal and CS, poor sleep quality, pain, anxiety, and emotional dysregulation, standing out. The few studies that have explored the presence of PTSSs/PTSD in groups with CSS, especially FM, have clearly shown a mutual exacerbation of symptoms, generating greater pain and somatization, emotional distress, dysfunctional emotional coping, and disability [18,28,29,32], and poor sleep quality [28,32]. The prevalence of PTSD in the CSS group (27.40%) was much higher than that reported in the general population, which is approximately 5% [22]. This concurs with previous reports that found a rate of PTSD of 10–20% in individuals with chronic pain, 35–40% in individuals with FM, and more than 50% in war veterans with pain [21,22,28].

Finally, based on previous studies and the pattern of correlations observed, sleep quality, pain catastrophizing, and emotional dysregulation were examined as mediators of the relationship between PTSD and pain intensity in the CSS subgroup with PTSD. We found that all three variables were significant mediators of the association between PTSD and pain, accounting for a considerable percentage of variance (55.3%). To the best of our knowledge, no previous studies have considered the variables included in our research. A previous work did not identify depression as a significant mediator between PTSD and FM [30]. Miró et al. highlighted the mediating role of anxiety in the association between PTSS and daily functioning in FM [32], which may be consistent with the present findings. Anxiety has been shown to have an especially strong correlation with pain catastrophizing [14,16], that also it is considered underlying cognitive process in emotional disturbances [13], and both are highly correlated with emotional dysregulation [14]. It has been shown that past trauma experiences can contribute to dysfunctional cognitive–affective and interpersonal characteristics, such as early maladaptive schemas and insecure attachment [34], which correlate with the tendency to catastrophize [14], and emotional dysregulation, which increases anxiety and negative affect [6,16,56]. In fact, pain catastrophizing has been correlated with hyperactivity in the anterior cingulate cortex, which is involved in emotional processing and pain modulation, as well as areas of the prefrontal cortex associated with cognitive appraisal and emotional regulation [57].

Poor emotional regulation and the tendency to catastrophize distress could perpetuate a hyperactivation state that contributes to the emergence of CS. This would support the perpetual avoidance model that suggests that dysfunctional cognitive processing following trauma, especially intrusions and catastrophizing, generates psychological hyperactivation and avoidance behaviors that maintain both trauma and pain [58]. This proposal is also consistent with the hypermnesia–hyperactivation model [59], which hypothesizes that traumas involving physical pain produce a type of memory trace, and the neuroendocrine hyperactivation associated with the trauma reinforces it since both problems share similar neuroanatomical areas [60]. Similarly, the recent imbalance in the threat and soothing systems model highlights that after suffering adversity, the “salience network” (midcingulo-insular network) remains in a continuously alert mode, and sustained hyperactivation contributes to CSS [35].

An obvious consequence of maintaining a state of hyperactivation is disturbing sleep quality [10,61], which plays an important role in the development and maintenance of PTSD [62] and may also contribute to the genesis of pain after trauma [63]. CSS patients show alterations in the architecture and microstructure of sleep that have been associated with a state of hyperarousal prior to sleep [8,9,10]. Both clinical and experimental studies have shown that altered sleep quality reduces pain tolerance and alters central pain inhibition [10,11,12]. Poor sleep quality also affects the HPA axis, which may result in a breakdown of the endocrine, sympathetic, and immune corporal systems, resulting in a CS state [2,10,60]. Maintaining dysfunctional emotional activation and the impossibility of recovering through sleep may perpetuate the state of hyperarousal, generating a vicious circle and allostatic overload that tends to maintain itself.

The present study has several limitations. First, cross-sectional designs have difficulties capturing the true nature of mediational processes; therefore, conclusions should be made with caution. Second, recall bias could be a limitation in retrospective studies, although this type of report tends to underestimate rather than overestimate trauma recall [23]. Apart from using questionnaires with adequate psychometric properties, an online survey helped minimize social desirability, and we used two suitable control groups to reduce confounders. Third, although we included a large sample of both sexes (*n* = 1542), the small percentage of men in the CSS group prevented sex-based analysis. Future research with longitudinal or multi-wave designs is needed to better substantiate the proposed mechanisms and strengthen the evidence regarding the causal order of the variables, as well as to incorporate diverse samples (gender, demographics, and sociocultural characteristics) and a broader range of objective and subjective measures.

## 5. Conclusions

In conclusion, a higher percentage of participants in the CSS and MP groups reported traumatic experiences (mainly emotional and physical trauma in adolescence or adulthood) compared to participants in the healthy group, although participants in the CSS group had a greater presence of repeated trauma, higher rates of PTSD, and PTSD with dissociative symptoms. The CSS group had greater clinical impairment than the other groups, especially the CSS subgroup with PTSD. In this last subgroup, sleep dysfunction, pain catastrophizing, and emotional dysregulation mediated the relationship between PTSD and pain. CSS remains a major therapeutic challenge. Evidence-based approaches to CSS, such as Cognitive Behavioral Therapy focused on pain (CBT-P) or sleep disturbance such as insomnia (CBT-I) and Acceptance and Commitment Therapy (ACT), have demonstrated efficacy and significant benefits compared to other active treatments, although the effect sizes of these therapies on pain intensity are moderate [64,65], and trauma is not addressed at all. Concerning PTSD, trauma-focused therapies, such as Prolonged Exposure, Cognitive Processing Therapy, Cognitive Therapy for PTSD, or CBT trauma-focused, are widely recognized and supported by solid evidence [66]. These approaches are considered first-line treatments for treating CSSs or PTSD separately; however, there are no clinical guidelines that recommend how to treat both conditions together. Preliminary studies indicate that treating CSS and PTSD simultaneously with Emotional Awareness and Expression Therapy (EAET) may provide greater clinical benefits for both conditions than usual approaches, such as CBT or ACT [56]. However, more research is needed. The present findings highlight the need to assess the presence of PTSSs/PTSD in people with CSSs and suggest that, in the population with both conditions, a treatment that addresses trauma and pain, with an emphasis on improving sleep quality, pain catastrophizing, and emotional regulation, could improve the effectiveness of the current therapeutic approach to these syndromes. Effective trauma and pain therapies should not be limited to CBT-P/CBT-I or ACT but should also incorporate interventions that address emotional dysregulation and pain with EAET or other body-oriented methods, which could impact central sensitization in the nervous system. This integrative biopsychosocial approach has the potential to enhance clinical outcomes and improve the quality of life of individuals suffering from both CSSs and PTSD.

## Figures and Tables

**Figure 1 healthcare-13-02221-f001:**
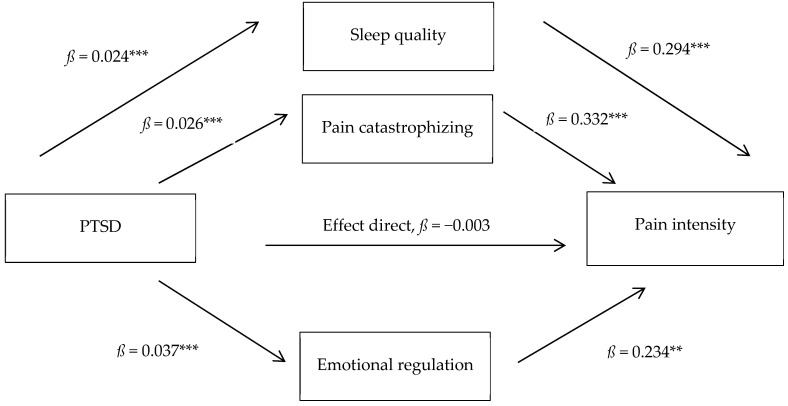
Multiple mediation model of the association between post-traumatic stress disorder (PTSD) and pain. Note: *ß* = non-standardized beta. ** *p* < 0.01; *** *p* < 0.001.

**Table 1 healthcare-13-02221-t001:** Clinical characteristics of the study groups.

	CSS (*n* = 467)		MP (*n* = 214)		Healthy (*n* = 861)		*F* (2,1539)	*ηp* ^2^
	*M*	*SD*	*M*	*SD*	*M*	*SD*		
Central sensitization	41.11	15.71	31.97	14.63	29.51	13.34	101.07 *** (a, b)	0.11
Pain intensity	4.10	2.42	3.09	2.22	2.47	1.77	96.22 *** (a, b, c)	0.11
Sensory–affective pain	8.11	8.81	4.55	6.59	3.28	4.51	86.78 *** (a, b, c)	0.10
Sleep quality	8.03	4.23	6.75	3.84	6.05	3.25	44.05 *** (a, b, c)	0.05
Anxiety	8.58	4.45	6.73	4.39	6.64	4.06	33.52 *** (a, b)	0.04
Depression	5.95	4.40	4.90	4.06	4.43	3.66	22.43 *** (a, b)	0.02
Pain catastrophizing	16.31	12.38	12.21	11.12	12.04	10.53	23.30 *** (a, b)	0.02
Perceived stress	25.83	9.29	22.62	10.50	24.31	8.87	9.44 *** (a, b)	0.01
Emotional regulation	4.33	1.98	3.68	1.95	3.77	1.87	15.28 *** (a, b)	0.01
Intrusion symptoms (PTSD)	5.25	4.05	4.13	3.84	3.77	3.55	12.61 *** (a, b)	0.03
Avoidance symptoms (PTSD)	2.07	1.85	1.75	1.87	1.55	1.79	6.45 ** (b)	0.01
Cognitive and mood symptoms (PTSD)	6.42	5.37	5.29	5.35	4.26	4.67	14.77*** (b)	0.03
Hyperarousal symptoms (PTSD)	6.00	4.53	4.46	4.09	3.57	3.62	28.94 *** (a, b)	0.06
Dissociative symptoms (PTSD)	1.20	1.61	0.81	1.51	0.72	1.32	8.77 *** (b)	0.02
Daily impairment (PTSD)	5.48	5.03	4.16	4.67	3.59	4.17	13.75 *** (a, b)	0.03

Note: CSSs = central sensitization syndromes, MP = medical pathologies, PTSD = post-traumatic stress disorder. a = significant differences between CSS group vs. MP group, b = significant differences between CSS group vs. healthy group, and c = significant differences between MP group vs. healthy group. ** *p* < 0.01; *** *p* < 0.001.

**Table 2 healthcare-13-02221-t002:** Comparison of trauma-exposed CSS participants who developed PTSD versus CSS participants who did not develop PTSD.

	CSS with PTSD (*n* = 128)		CSS without PTSD (*n* = 127)		*t* (253)	*d*
	*M*	*SD*	*M*	*SD*		
Central sensitization	49.90	15.56	34.86	13.37	8.27 ***	1.03
Pain intensity	4.83	2.22	4.06	2.46	2.60 **	0.32
Sensory–affective pain	12.22	10.54	6.02	7.01	5.52 ***	0.69
Sleep quality	9.49	4.13	6.75	4.03	5.34 ***	0.67
Anxiety	10.93	4.04	7.02	3.83	7.91 ***	0.99
Depression	7.52	4.38	4.39	3.82	6.08 ***	0.76
Pain catastrophizing	20.77	12.12	13.80	11.18	4.77 ***	0.59
Perceived stress	30.03	8.26	22.38	9.55	6.84 ***	0.85
Emotional regulation	5.06	1.58	3.62	1.97	6.38 ***	0.80

Note: CSSs = central sensitization syndromes. ** *p* < 0.01; *** *p* < 0.001.

**Table 3 healthcare-13-02221-t003:** Relationship between variables in the CSS subgroup with PTSD (*n* = 128).

	CSI	MPQ-SF- VAS	MPQ-SF-SA	PSQI	HADS-A	HADS-D	PCS	PSS	EPS-25
Intrusion symptoms (PTSD)	0.26	0.15	0.15	0.19	0.36 *	0.14	0.24	0.28	0.25
Avoidance symptoms (PTSD)	0.20	0.22	0.02	0.08	0.22	0.12	0.15	0.15	0.11
Cognitive and mood symptoms (PTSD)	0.38 *	0.22 *	0.16	0.22	0.48 *	0.31 *	0.31 *	0.40 *	0.43 *
Hyperarousal symptoms (PTSD)	0.60 *	0.42 *	0.46 *	0.47 *	0.53 *	0.39 *	0.47 *	0.49 *	0.53 *
Dissociative symptoms (PTSD)	0.36 *	0.25	0.23	0.17	0.38 *	0.32 *	0.20	0.31 *	0.43 *
Daily impairment (PTSD)	0.37 *	0.21	0.18	0.27	0.37 *	0.42 *	0.24	0.33 *	0.47 *

Note: CSSs = central sensitization syndromes, CSI = Central Sensitization Inventory, MPQ-SF = McGill Pain Questionnaire-Short Form (VAS = Visual Analog Scale, SA = Sensory–Affective Scale), PSQI = Pittsburgh Sleep Quality Index, HADS = Hospital Anxiety and Depression Scale (A = anxiety subscale, D = depression subscale), PCS = Pain Catastrophizing Scale, PSS = Perceived Stress Scale, EPS-25 = Emotional Processing Scale-25. Bonferroni Adjustment, * *p* < 0.0005.

**Table 4 healthcare-13-02221-t004:** Mediators of the effect of post-traumatic stress disorder on pain intensity.

Bootstrapping					
		Product of Coefficient			
	Point Estimate	*SE*	*z*	BC 95% CI	
				Lower	Upper
Direct effect	−0.003	0.006	−0.48	−0.013	0.010
Indirect effects					
Total	0.024	0.005	5.07 ***	0.017	0.033
Sleep quality	0.007	0.003	2.80 **	0.003	0.013
Pain catastrophizing	0.009	0.03	3.13 **	0.004	0.017
Emotional regulation	0.009	0.04	2.60 **	0.002	0.015
Total effect	0.022	0.006	3.72 ***	0.010	0.034

Note: *SE* = standard error, BC = bias corrected, CI = confidence interval. ** *p* < 0.01; *** *p* < 0.001.

## Data Availability

The datasets generated in this study are available from the corresponding author upon request.

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
