# Peer review of "Central Sensitization Syndromes and Trauma: Mediating Role of Sleep Quality, Pain Catastrophizing, and Emotional Dysregulation Between Post-Traumatic Stress Disorder and Pain"

_healthcare, 2025, doi:10.3390/healthcare13172221_

Round 1
Reviewer 1 Report
Comments and Suggestions for Authors
Strengths
The manuscript addresses a relevant and clinically important topic: the relationship between PTSD and pain in central sensitization syndromes (CSS), and the mediating role of sleep quality, pain catastrophizing, and emotional dysregulation.
The sample size (N=1542) is robust, the clinical measures are validated, and the statistical analyses (ANOVA, t-tests, multiple mediation with bootstrapped CIs, Bonferroni corrections) meet international standards.
Reporting effect sizes (Cohen’s d, ηp²) improves the clinical interpretability of the findings.
Major Comments
- Mediation model and causal inference
The mediation analysis is based on cross-sectional data, which limits causal conclusions regarding the PTSD → sleep/emotion variables → pain pathway. This is not a fatal flaw but should be explicitly acknowledged in the discussion, emphasizing that longitudinal or multi-wave designs would better substantiate the proposed mechanisms. - PTSD measurement (lack of CAPS-5)
Although the DSM-5-based, study-specific interview is appropriate, the authors should clarify why they did not use a standardized, internationally recognized diagnostic tool (e.g., CAPS-5 or PCL-5). This is important for external validity and cross-study comparability. - Detailed trauma subtype analysis
While several trauma types (psychological, physical, sexual, loss) are reported, their distinct contributions to clinical outcomes are not analyzed separately. Discussing whether different trauma types differentially influence the PTSD–pain relationship could provide more nuanced clinical insights. - Population context
Prior Spanish findings are briefly mentioned, but more detailed comparison of the observed PTSD and trauma prevalence rates with other Spanish or European samples would strengthen the discussion and contextualize the findings.
Minor Comments
- Minor grammatical and editorial issues should be corrected (e.g., “is is more clinically affected,” around line 114).
- Reference and DOI formatting is inconsistent in places and should be standardized before publication.
- The therapeutic recommendations, while valuable, remain somewhat general. Briefly citing evidence-based approaches (e.g., CBT-I for sleep, trauma-focused CBT, ACT) would make the clinical implications more actionable.
Reviewer 2 Report
Comments and Suggestions for Authors
General comments
I would like to begin by congratulating the authors on their important and timely contribution to the field with their manuscript titled “Central sensitization syndromes and trauma: mediating role of sleep quality, pain catastrophizing and emotional dysregulation between post-traumatic stress disorder and pain.”
This study addresses a complex and critically under-recognized intersection between trauma, traumatic loss and chronic pain mechanisms, particularly central sensitization syndromes (CSS), which remain largely unfamiliar territory for many trauma-focused practitioners and clinicians.
By exploring the mediating roles of sleep disturbances, pain catastrophizing, and emotional dysregulation in the relationship between post-traumatic stress disorder (PTSD) and chronic pain, the authors offer valuable insights into how psychological and physiological processes are deeply interconnected in trauma survivors. The findings strongly suggest that effective trauma therapy should not be limited to cognitive or narrative approaches alone, but should also incorporate body-oriented and somatic methods that directly address pain and dysregulation in the nervous system. While this last point remains a bit untreated in the discussion section of this manuscript.
This manuscript makes a compelling case for a more integrative biopsychosocial approach in trauma care—one that could significantly enhance clinical outcomes and improve the quality of life for individuals suffering from both PTSD and chronic pain conditions. I recommend the authors for shedding light on this often-overlooked dimension of trauma treatment, more explicitly and maybe by making the link to certain therapy methods (somatic experiencing, sensorimotor trauma therapy, etc.).
Participants
I would expect some explanation about the large size difference between the healthy group, the CSS group and the MP group.
How have the participants been recruited? Just through announcement on the internet, social media, etc.? What have the researchers tried/done in order to motivate participants to enroll in this study? What have been there attempts in order to balance the participation of men and women?
Measures
Self-report scales can provide information about symptoms which are indicative for PTSD but not for PTSD as a clinical diagnosis (diagnosis is only possible through the use of a semi-structured clinical interview such as the CAPS). This should appear more clearly in the text.
The self-developed interview to assess the presence of trauma, PTSD/PTSS. No information is offered on its psychometric qualities. Why did the authors decide not to use a validated (trauma) scale (such as the PCL-5)? Why did they decide not to use a 5-point Likert Scale?
How were the scores on this scale (not at all, a little, quite a lot, a lot) computed (into a dichotomic variable, needed for the presence of symptoms indicative for PTSD or not)?
No information on the cut-off scores used for this variable?
Other instruments are better explained. I would also like some more information on their use in previous studies (with Spanish participants).
I would include more information on the psychometric qualities of the instruments used to measure the important variables of this study.
For example: The Central Sensitization Inventory (CSI) demonstrates strong psychometric properties, including good reliability and validity, making it a useful tool for identifying individuals with central sensitization in chronic pain populations. Specifically, the CSI shows strong test-retest reliability, internal consistency, construct validity, criterion validity, and cross-cultural validity.
See: Pain Pract.. 2022 Sep 22;22(8):702–710. doi: 10.1111/papr.13162
Validity of the Central Sensitization Inventory compared with traditional measures of disease severity in fibromyalgia
Fausto Salaffi, Sonia Farah, Claudia Mariani, Piercarlo Sarzi‐Puttini, Marco Di Carlo
PMCID: PMC9826291 PMID: 36097821
Some more information on previous research with Spanish participants that used the same scales would also be usefull.
Generally, I like the detailed description of statistical methods and analyses. That part was much appreciated.
Results
It would have been interesting to explore the effect of the type of trauma on the type of symptoms since the authors also discuss the role of physical/body symptoms and hyperarousal in the development/perception of pain.
- Discussion
In this section, I read most language/spelling errors. That indicates that this manuscript needs an extra thorough proof reading!
Some examples of TYPOS:
Line 368. Previous studies have highlightED that experience traumatic events… something is wrong with this sentence
Line 389. After emotional trauma previous studies have also stand out… (what does this mean: have stand out?)
Line 396. Furthermore, although several studies have (instead of has)
Line 444. … that maintain (instead of maintains)
Line 449. … that after sufferING instead of … that after suffer
Finally, I would appreciate an extra paragraph in the discussion section about the clinical implications of the results of this study for the treatment of CSS/PTSD patients.
Round 2
Reviewer 1 Report
Comments and Suggestions for Authors
Dear Authors,
Thank you very much for your detailed and precise responses as well as the thoughtful modifications to the manuscript. I am satisfied with the revisions and hereby recommend the study for publication.
I wish you continued success in your future work.
Sincerely,
Reviewer
Author Response
Dear Authors,
Thank you very much for your detailed and precise responses as well as the thoughtful modifications to the manuscript. I am satisfied with the revisions and hereby recommend the study for publication.
I wish you continued success in your future work.
Sincerely,
Reviewer
Dear Reviewer,
Thank you so much for your reply and your well wishes for our future research!
Sincerely,
The authors
Reviewer 2 Report
Comments and Suggestions for Authors
I did not review the whole manuscript. I stopped reading after the 'Introduction' because I find it unacceptable that there are still so many layout and language issues in a second review.
I am willing to review this manuscript again and invest the necessary amount of time after the necessary corrections in spelling and layout but this is not acceptable for me.
Just a sample of problems:
Line 24 - delete extra space
Line 37 - insert space in 'demonstratedto'
Line 77 - insert space in 'highlightedthat'
Line 105 - Miro et al. found that anxiety play... should be 'plays'
Line 111 - Thisstudy
Round 3
Reviewer 2 Report
Comments and Suggestions for Authors
Final Review Report
I have now reviewed the revised version of the manuscript entitled “Central Sensitization Syndromes and Trauma: Mediating Role of Sleep Quality, Pain Catastrophizing, and Emotional Dysregulation Between Post-Traumatic Stress Disorder and Pain” for the second time.
The authors have submitted a significantly improved and well-clarified version of their work, along with a thorough and constructive cover letter that carefully explains how the manuscript was revised in response to previous feedback. I would like to sincerely thank and congratulate the authors for the substantial effort and thoughtful attention they have invested in strengthening the quality and clarity of their article. Their engagement with the review process is commendable.
I now recommend the publication of this article in its current form, subject to a final and careful proofreading for language, spelling, and formatting errors. This has already been requested in earlier rounds, but a few minor issues still remain in the current version. These include:
-
Line 18: Insert space between "trauma" & "physical"
-
Line 52: "ofdescending" – insert space
-
Line 119: "greaterclinical" – insert space
-
Line 156: Change "is" to "was" → “Our final interview was…”
-
Line 164: "We add questions in which..."
-
Line 183: "We included items that..."
-
Please ensure consistent use of past or present tense throughout the manuscript (preferably past tense for reporting completed methods and results).
-
-
Line 183: Use correct spelling → “subscales” instead of “subescales”
These are relatively minor corrections but important to address for ensuring professional polish and readability. I trust that these can be resolved quickly in the final proofing stage.
Thank you again to the authors for their valuable contribution to the field and for the considerable care they’ve shown in responding to prior feedback. I look forward to seeing this work published.
Recommendation: Accept with minor final language and formatting corrections